# Low LEF1 expression is a biomarker of early T-cell precursor, an aggressive subtype of T-cell lymphoblastic leukemia

Mei Wang[1]*, Chi Zhang[2]

1 School of Life Science, Peking University, Beijing, People's Republic of China, 2 Institute of Vertebrate Paleontology and Paleoanthropology, Chinese Academy of Sciences, Beijing, People's Republic of China

* mei_wang@outlook.com

**Data Availability Statement:** All the data in this study is available in public domains. The URL links of the public datasets (GSE42328 and TARGET T-ALL) are as follows: a.TARGET T-ALL: https://ocg.cancer.gov/programs/target, dbGaP ID:

## Abstract

Early T-cell precursor (ETP) is the only subtype of acute T-cell lymphoblastic leukemia (T-ALL) listed in the World Health Organization (WHO) classification of myeloid neoplasms and acute leukemia. Patients with ETP tend to have worse disease outcomes. ETP is defined by a series of immune markers. The diagnosis of ETP status can be vague due to the limitation of the current measurement. In this study, we performed unsupervised clustering and supervised prediction to investigate whether a molecular biomarker can be used to identify the ETP status in order to stratify risk groups. We found that the ETP status can be predicted by the expression level of Lymphoid enhancer binding factor 1 (LEF1) with high accuracy (AUC of ROC = 0.957 and 0.933 in two T-ALL cohorts). The patients with ETP subtype have a lower level of LEF1 comparing to the those without ETP. We suggest that incorporating the biomarker LEF1 with traditional immune-phenotyping will improve the diagnosis of ETP.

## Introduction

Early T-cell precursor (ETP) is the only subtype of acute T-cell lymphoblastic leukemia (T-ALL) listed in the World Health Organization (WHO) classification of myeloid neoplasms and acute leukemia [1]. It is defined by immune-phenotyping as lack of CD1a and CD8, weak expression of CD5 and positive for one or more of the myeloid/stem cell markers [2]. The unique phenotype represents the characteristics of immature T-cells which correspond to the early stage of normal T-cell development. Contrary to "matured" T-ALLs, patients with ETP have worse disease outcomes [2–5]. Nowadays, most of the T-ALLs can be treated. However, the relapse rate is observed as up to 40% [6]. Targeting the specific ETP sub-group which has a higher relapse rate is crucial for treating T-ALLs.

As one of the routine examinations, the expression of relevant cell surface markers was measured by the flow cytometry [7]. However, the diagnosis can be varied depending on different pathologists and/or different parameters, for instance, the voltage setting. In order to improve the reliability of the diagnosis, Khogeer et al. developed a scoring system based on 11 surface markers to re-define the ETP subgroup [7]. Still, it is solely based on the measurement

phs000464 b.GSE42328: https://www.ncbi.nlm.nih.gov/geo/ c.TCGA PanCancer: UCSC Xena platform (https://pancanatlas.xenahubs.net) Additionally, the authors have provided the Scripts data as a Supporting Information file.

**Funding:** This research is supported by National Natural Science Foundation of China (81900155) to M.W., the 100 Young Talents Program of Chinese Academy of Sciences and the Strategic Priority Research Program of Chinese Academy of Sciences (XDB26000000) to C.Z.

**Competing interests:** None of the authors has a relevant conflict of interest.

of flow cytometry. In this study, we investigate whether a molecular biomarker can be used to identify the ETP status in order to stratify risk groups.

## Results and discussion

### Inconsistent classification of ETP status yield by different classifiers

Currently, the most well accepted classification of ETP status was defined by immune pheno-type proposed by Coustan-Smith et al. in 2009 [2]. According to their study, the sub-group of patients was discovered by unsupervised clustering using 35 differentially expressed genes in mice thymic early T-cell precursors. Patients in that group manifested a unique molecular character. Since then, this immune phenotype has been accepted as the definition of ETP [8]. How the ETP subtype was discovered indicates that the classification by immune-phenotyping and clustering by gene expression profile should reach an agreement. To evaluate if and to what extent the previously reported clustering methods could identify ETP status, we performed unsupervised clustering in two T-ALL datasets (TARGET T-ALL [9] and GSE42328 [10]). The characteristics of the two datasets were described in Materials and Method below.

Clustering by the whole transcriptome and by the 35-gene ETP signature were conducted on the two T-ALL cohorts (TARGET and GSE42328). Clustering based on the whole transcriptome is aiming to investigate whether distinct subtypes with molecular characteristics exist, and clustering based on the 35 ETP signature genes is to identify the ETP group in patients [2]. The subjects were divided into two clusters, but inconsistent classifications were yield by different approaches (Fig 1). Approximately 10–20% of T-ALLs were classified as ETP (n = 19, 10.05% in TARGET, n = 10, 18.87% in GSE42328) based on the immune-phenotype. However, more than half of the samples were predicted as ETP (n = 100, 60.61% in TARGET; n = 32, 60.38% in GSE42328) when clustering using the whole transcriptome, while 62 patients (37.58%) were predicted as ETP in TARGET and 34 (64.15%) in GSE32428 when clustering using the 35-gene panel. Thus, the classification of ETP by immune-phenotyping is inconsistent with that by unsupervised clustering of gene expression profile. which deviates from the original finding [8]. It indicates that the ETP status cannot be correctly classified by unsupervised clustering approaches using either the whole transcriptome or the 35-gene ETP signature.

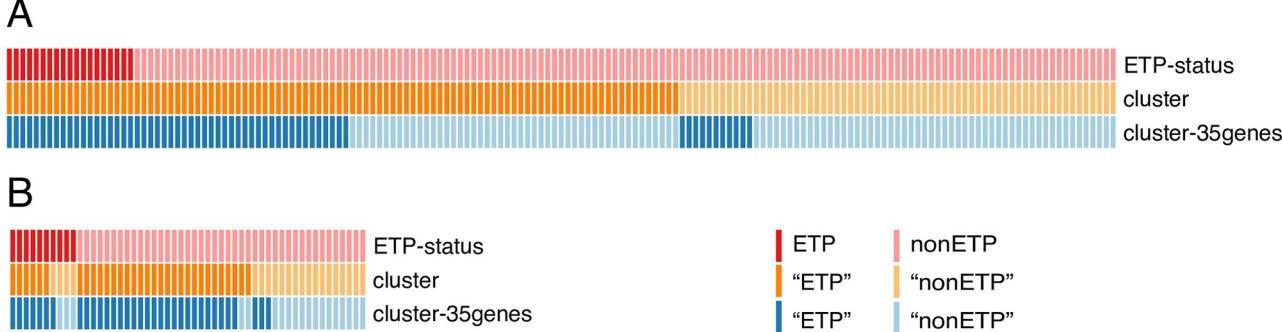

**Fig 1. Classifications of ETP by immune-phenotype, transcriptome, and 35-gene ETP signature genes in TARGET T-ALL (A) and GSE42328 (B) datasets.** Each column is a patient and each row represents a classification method. Patients were stratified to ETP (dark color) or nonETP (light color) groups based on immune phenotyping (first row), the whole transcriptome clustering (second row) or the 35-gene ETP signature clustering (third row), respectively.

## A potential biomarker of ETP status—LEF1

We then developed a lasso regression model to evaluate whether the ETP status could be predicted correctly based on transcriptomic data. The prediction accuracy of the model is first-rate that the area under the curve (AUC) of the receiver operating characteristic (ROC) achieved 0.957 (confidence interval (CI): 0.946–0.969, Fig 2A). It is concluded that the immune-phenotype defined ETP status can be predicted by gene expression profile.

To investigate which gene and to what extent it contributes to the prediction model, we looked into the features which were selected in the 100 rounds of outer cross-validation. The top 10 predicters were listed in Fig 2B. The coefficient of LEF1, CD5, HOXC10, and RSPO4 were high comparing to the other top predictors (Fig 2C). To predict the ETP status by expression level of individual genes, LEF1, CD5 and OGN achieved high accuracy in terms of AUC of ROC (Fig 2D).

Lymphoid enhancer binding factor 1 (LEF1) is the top player, which was selected 94 times out of 100 (Fig 2B). LEF1 is one of the most contributed variables. The coefficient of LEF1 ranged from –0.826 to –0.012 within the 94 times (Fig 2C). This gene is an important transcription factor in T-cell development and malignancy [11, 12]. It is also one of the most differentially expressed genes (DEGs) between ETP and nonETP in TARGET T-ALL dataset (S1 Table). There were 8761 genes out of 33038 differentially expressed and LEF1 was ranked 12th in the DEGs. Patients with ETP have a lower expression of LEF1 comparing to patients with "matured" T-cells.

We further tested whether LEF1 could predict the ETP status in the GSE42328 dataset. The prediction of ETP status by LEF1 expression level reached 0.933 (CI: 0.858–1, Fig 2E). The expression level of LEF1 in patients with early immature subtype is significantly lower than the cortical/mature group (Fig 2F). LEF1 was also found differentially expressed between ETP and nonETP subtypes in two previous Microarray datasets (GSE28703 and GSE8879) [2, 13].

To quantify the prediction accuracy in terms of sensitivity and specificity, we considered the identification of ETP by immune-phenotyping as the golden standard. The sensitivity (0.9 in TARGET, 1 in GSE42328) and specificity (1 in TARGET and 0.8 in GSE42328) of the prediction by LEF1 expression level are higher than either by the whole transcriptome clustering or by the 35-gene signature clustering in both TARGET T-ALL and GSE42328 (Table 1).

It is known that LEF1 is essential for T- and B-cell differentiation and lineage determination [14–16]. Yu et al. demonstrated that Tcf1 and Lef1 transcription factors are intrinsic required in leukemic stem cells (LSCs) self-renewal [17]. It indicates that LEF1 plays a key role in T-ALL tumorigenesis and provides an important biological evidence to support our hypothesis that expression level of LEF1 could predict the ETP status. In this study, the high-risk subtype of T-ALL showed a lower expression of LEF1. Jia et al. also found that LEF1 expression is a favorable prognostic factor in a pediatric cohort with 94 B-ALLs and 28 T-ALLs [18]. To the contrast, the expression of LEF1 in solid tumors is higher than the corresponding normal tissue. The overexpression of LEF1 was observed in most of the solid tumors (15 out of 24 types in the TCGA Pancancer database, S2 Table). Moreover, overexpression of LEF1 was associated with worse overall survival in adrenocortical cancer, kidney clear cell carcinoma, kidney papillary cell carcinoma, rectum adenocarcinoma, and uveal melanoma (S3 Table). However, in thymoma, a negative association with the LEF1 expression level and overall survival rate was observed. It is interesting that thymus is the organ where the development of T- and B-cell takes place (S3 Table). Giambra et al. reported that the activity of leukemia stem cells is dependent on Lef1 in the Wnt-active subpopulation of T-ALL [19]. In addition, it has also been found that low expression of LEF1 is associated with worse survival in acute myeloid leukemia

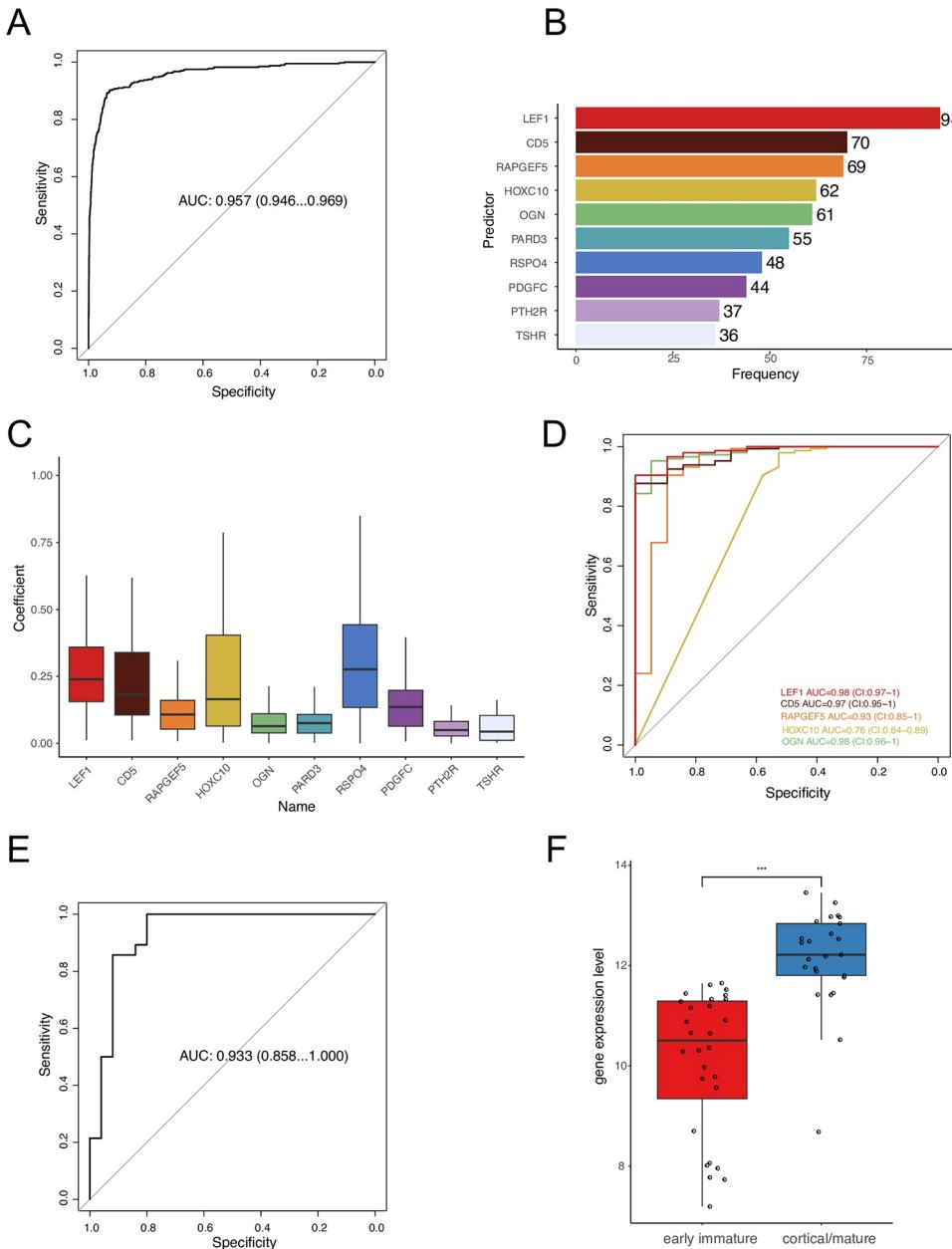

**Fig 2. (A)** AUC of ROC, predictions based on transcriptome against ETP status in the TARGET T-ALL dataset; **(B)** The top 10 predictors with non-zero coefficients in the 100 rounds of outer CV; **(C)** The coefficients of the top 10 predictors in the 100 rounds of outer CV. The y-axis is the absolute value of coefficient; **(D)** AUC of ROC, individual-gene expression against ETP status in the TARGET T-ALL dataset; **(E)** AUC of ROC, LEF1 expression against T-ALL subtypes (early immature, cortical/mature) in the GSE42328 dataset; **(F)** the expression level of LEF1 in T-ALL subtypes in the GSE42328 dataset. AUC, the area under the curve; ROC, the receiver operating characteristic; ETP, early T-cell precursor.

**Table 1. The accuracy (sensitivity and specificity) of predicting the ETP status (defined by immune markers).**

|  |  | Sensitivity | Specificity |
|---|---|---|---|
| TARGET | cluster | 1 | 0.45 |
|  | 35-gene cluster | 1 | 0.71 |
|  | LEF1 | 0.9 | 1 |
| GSE42328 | cluster | 0.6 | 0.4 |
|  | 35-gene cluster | 0.3 | 0.6 |
|  | LEF1 | 1 | 0.8 |

[20]. Overall, it suggests that LEF1 plays different roles in solid tumorigenesis and lymphoid malignancies. The underlying mechanism needs further investigation.

## Conclusions

In this study, we investigated the molecular characteristics of the risky subtype of T-ALL–ETP. The current routine diagnosis of ETP is vague. We found that the ETP status can be predicted by the expression level of LEF1 with high accuracy. We propose that incorporating the biomarker LEF1 with traditional immune-phenotyping will improve the diagnosis of ETP.

## Materials and methods

### Study subjects

The TARGET T-ALL (n = 165, children and adolescent) dataset was used as the training set to develop the prediction model. Gene expression and clinical data were retrieved from Therapeutically Applicable Research To Generate Effective Treatments (TARGET, https://ocg.cancer.gov/programs/target, dbGaP ID: phs000464). The complete data of TARGET T-ALL in the original study were composed of 264 children and young adults diagnosed with T-ALL [9]. 190 cases had immune phenotypes in terms of ETP status, 25 out of which were labelled as "nearETP" and excluded due to the ambiguous definition, resulting in 165 cases used in this study. We compared the demographic and clinical characteristics of the selected samples to those in the original publication (Table 2). There are 33,038 genes totally in our dataset. The gene expression data were normalized using Trimmed Mean of M-values (TMM) [21] and log2 transformed.

The GSE42328 adult T-ALL (n = 53) was the validation dataset. The GSE42328 dataset had 53 adult T-ALLs. The clinical characteristics of the samples were well described in their

**Table 2. The demographic and basic clinical characteristics of the samples selected from TARGET T-ALL.**

|  | Original study (n = 265) | Used this study (n = 165) |
|---|---|---|
| Age (Median, range) | 9 (1–29) | 9 (1–22) |
| Gender (male, n, %) | 202 (76.23%) | 122 (73.94%) |
| WBC (mean) | 167.54 | 163.3 |
| Bone marrow blasts at diagnosis (mean) | 91.56 | 91.51 |
| ETP status |  |  |
| ETP (n) | 19 | 19 |
| nonETP (n) | 146 | 146 |
| nearETP (n) | 25 | NA |
| Unknown (n) | 75 | NA |

original study [10]. Data was downloaded from Gene Expression Omnibus (GEO, https://www.ncbi.nlm.nih.gov/geo/). In total, 47,212 probes were measured representing 31324 genes. The value of gene expression provided were quantile normalized and log2 transformed.

The TCGA pan-cancer dataset was used to investigate the association of the expression level of LEF1 with survival in 32 types of tumor. Data were downloaded from Pan-Cancer Atlas Hub at UCSC Xena platform (https://pancanatlas.xenahubs.net) [22]. In total, 9110 samples were selected with sample type labeled as primary tumor. The RNA sequencing data were batch-effects normalized and log2 transformed.

All the data in this study are available in public domains. Ethical approval is not applicable for this study.

## Unsupervised clustering

Clustering by the whole transcriptome and by the 35-gene ETP signature were conducted respectively on the two T-ALL cohorts (TARGET and GSE42328). For the clustering by the whole 75 transcriptome, the top 5,000 most variable genes were selected [23]. For the clustering by 35-gene ETP signature, only the 35 genes (or the corresponding probes) were selected [2]. Hierarchy clustering with Pearson's correlation coefficient was performed 10,000 times to achieve a consensus [24]. The procedure was conducted by R package ConsensusClusterPlus v1.48 [23].

## Supervised prediction

A regression model that uses L1-regularization (lasso)[25] was applied to fit the data and to predict the ETP status by transcriptomic gene expression data (RNA sequencing). Nested cross validation (CV) was conducted to evaluate the model performance. For outer CV, 80% of the patients (132 out of 165) were randomly assigned to the training set and the rest 20% (33 out of 165) to the test set. The outer CV was run 100 times. For inner CV, 5-fold CV was performed to optimize the tuning parameter lambda. The TARGET T-ALL (n = 165) dataset was used to develop the prediction model and GSE42328 adult T-ALL (n = 53) was the validation dataset. The prediction was conducted by R package glmnet v2.0 [26].

Receiver operating characteristic (ROC) curves of prediction against true ETP status were constructed. The area under the ROC curve (AUC) and 95% confidence interval (CI) were generated to compare model performance. The AUCs of ROC curves were compared by the DeLong test [27]. The decision boundary was determined at the point closest to the top-left part of the ROC curve using the R package pROC v1.15 [28]. Individuals were predicted as ETP if their predicted probability was larger than or equal to the cut-off point; otherwise, they were classified as nonETP.

**Other statistics.** Gene expression data (RNA sequencing) were compared between with ETP and without ETP to determine differentially expressed genes (DEGs). Differential expression analysis was conducted by R package DESeq2 v1.24 [29]. To test if LEF1 was abnormally expressed in tumor, nonparametric Wilcoxon two-group comparison was conducted in each tumor type in the TCGA Pan-Cancer dataset. Cox regression [30] was used to test the association between LEF1's expression level and overall survival in each tumor type.

To quantify the prediction accuracy in terms of sensitivity and specificity, we considered the identification of ETP by immune-phenotyping as the golden standard. Sensitivity is calculated as the proportion of patients with immune positive that are correctly identified by other methods (clustering or LEF1 expression level). Specificity is calculated as the proportion of patients with immune negative that are correctly identified as such.

## Supporting information

**S1 Table. Differential expression analysis on ETP status in TARGET T-ALL.**
(XLSX)

**S2 Table. Association analysis between LEF1 expression level and cancer in TCGA Pancancer dataset.**
(XLSX)

**S3 Table. Survival analysis on LEF1 expression level and patients' overall survival in TCGA Pancancer dataset.**
(XLSX)

**S1 File.**
(ZIP)

## Author Contributions

**Conceptualization:** Mei Wang, Chi Zhang.

**Data curation:** Mei Wang.

**Formal analysis:** Mei Wang.

**Funding acquisition:** Mei Wang, Chi Zhang.

**Investigation:** Mei Wang.

**Methodology:** Mei Wang.

**Project administration:** Mei Wang.

**Visualization:** Chi Zhang.

**Writing – original draft:** Mei Wang.

**Writing – review & editing:** Mei Wang, Chi Zhang.

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
