## [Decision Letter · Decision Letter 0]

30 Jan 2020

PONE-D-20-01174

low LEF1 expression is a biomarker of early T-cell precursor, an aggressive subtype of T-cell lymphoblastic leukemia

PLOS ONE

Dear Dr. Wang,

Thank you for submitting your manuscript to PLOS ONE. After careful consideration, we feel that it has merit but does not fully meet PLOS ONE’s publication criteria as it currently stands. Therefore, we invite you to submit a revised version of the manuscript that addresses the points raised during the review process by both Reviewers, experts in the field.

We would appreciate receiving your revised manuscript by Mar 14 2020 11:59PM. To enhance the reproducibility of your results, we recommend that if applicable you deposit your laboratory protocols in protocols.io, where a protocol can be assigned its own identifier (DOI) such that it can be cited independently in the future. For instructions see: http://journals.plos.org/plosone/s/submission-guidelines#loc-laboratory-protocols

We look forward to receiving your revised manuscript.

Kind regards,

Francesco Bertolini, MD, PhD

Academic Editor

PLOS ONE

2. At this time, we ask that you please provide the specific URL links of the public datasets, GSE42328 and TARGET T-ALL, used in your study.

Reviewers' comments:

Reviewer's Responses to Questions

**Comments to the Author**

1. Is the manuscript technically sound, and do the data support the conclusions?

Reviewer #1: Partly

Reviewer #2: Partly

2. Has the statistical analysis been performed appropriately and rigorously? 

Reviewer #1: I Don't Know

Reviewer #2: Yes

3. Have the authors made all data underlying the findings in their manuscript fully available?

Reviewer #1: Yes

Reviewer #2: Yes

4. Is the manuscript presented in an intelligible fashion and written in standard English?

Reviewer #1: No

Reviewer #2: Yes

5. Review Comments to the Author

Reviewer #1: Wang and Zhang report in silicon data pointing towards the differential diagnostic potential of LEF1 expression to detect early precursor T-ALL, in which it is expected to be reduced.

First of all the paper suffers from the at many occasions poor and difficult to understand written English that needs editing.

Second, since the paper will primarily be percepted by MD who are not that aware of the biostatistical methods used in silicon, proper citation of papers describing what exactly has been done and its feasibility should be considered.

I am not sure that statements as this given on lines 94-95 are per se deducible from the databases used and are justified.

Many additional statements are poorly supported by numeric data in the current paper version. Figure 1 is not telling a lot without more explanatory notes in the legend, actually I was not able to link the data on lines 108-112 to the optics of the figure.

Finally, the results need tissue validation on 5 biopsies of early precursor T-ALL in comparison with 10 more "maturated" cases by means of immunohistochemistry with the nicely working and several times reported antibody.

The discussion on the role of LEF1 in tumourigenesis does not add anything to the message of the paper and may be omitted.

Reviewer #2: Wang and Zhang in this research article investigated, through a clustering approach, if different methods used to diagnose Early T-cell Precursor (ETP) on Acute T-Cell Lymphoblastic Leukemia (T-ALL) are consistent between them. Moreover, they applied a prediction model in order to validate the goodness of LEF1 as molecular biomarker in detecting the status of ETP and therefore stratifying a risk group, moreover they used this approach to find other molecular biomarkers that might be useful to improve the diagnosis of ETP.

They found that the methods used to diagnose ETP not always overlap between them and that LEF1 marker has high accuracy in explaining the ETP status, finally they produced interesting and comparable results to LEF1 using other biomarkers.

The purposes of the study are interesting and well explained, but I was wondering if the authors could give a better explanation I) of the study subject used II) of the computational methods used, and III) of the results obtained. These three points are fundamental to make the study suitable for publication because they will manifest the accuracy of the methodology used and they will give the tools to reproduce the analyses and to understand them.

For this reason I would not consider the paper in its present form suitable for publication, but it shows sufficient potential to be reconsidered if it will be substantial revised in the major and minor points as I indicated below.

Major points:

I am suggesting a substantial extension of the method section with a better explanation of the parameters used and the reason why the authors opted for these specific unsupervised clustering and prediction model. I would also suggest to use, beside the one already used, another unsupervised clustering method to validate the results obtained.

Additionally I found some results not properly explained and improvable with some changes in the text and in the figures.

Specifically,

- In the study subjects section, the authors refer to a TARGET T-ALL dataset which includes 165 samples. I suggest to explain the characteristics of this dataset and why the authors preferred this data among the others available. In order to do this the author can report the website and the definition of the acronym already at the beginning of this section.

Additionally, looking at the TARGET publication guidelines of this dataset (https://ocg.cancer.gov/programs/target/target-publication-guidelines) the “TARGET data are available without restrictions on their use in publications or presentations, with the exception of the integrated Acute Lymphoblastic Leukemia (ALL) dataset.” and the “Investigators may only publish an ALL manuscript before the TARGET project team has published their global analysis on that tumor type IF the publication uses a very limited dataset (less than 5 genes) or the author has received written approval from the appropriate TARGET disease project team leaders.”. At this point, since this study is focused on the ALL dataset I was wondering if either this study used only 5 genes or the authors obtained the written approval to use this data. Unfortunately both of these information are missing in the current version of the manuscript.

Moreover, in this study 53 individuals has been taken from GEO with the ID GSE42328 but the reference from where these data have been generated which is “Van Vlierberghe P, Ambesi-Impiombato A, De Keersmaecker K, Hadler M et al. Prognostic relevance of integrated genetic profiling in adult T-cell acute lymphoblastic leukemia. Blood 2013 Jul 4;122(1):74-82. PMID: 23687089”, is not indicated, please cite it and add it to the bibliography.

Finally, please report the number of genes (total, median, mean of each of the dataset) and be more specific about the normalization and logarithmic transformation performed.

- In the unsupervised clustering method section it is not reported how the selection of the top 5,000 most variable genes occurred, please add also the reason why the author chose these number, a reference might help.

Additionally, please add a reference that justifies the use of Pearson’s correlation coefficient for “10,000 times to achieve a consensus” or extend this part in order to give the reader a better explanation of the methods employed. As indicated above I also suggest to apply another clustering method to confirm these results.

- In the prediction model section the authors report that the “80% of the patients were randomly assigned to the training set and rest 20% to the test set”, I find disagreement with the numbers indicated in the previous sections: 53 (test-set in the study subjects section) is not the 20% of 218 (165 of TARGET + 53 of GSE4328); moreover, even if it was, the 20% were not randomly chosen because they belong to the same group/experiment. Please, clarify this and include references and a better explanation of the parameters used in here.

- Line 124-125 I see from the plot that the two highest accuracy values in terms of AUC of ROC are LEF1 and OGN not LE1 and CD5 as indicated in the manuscript, please, clarify this part.

- Line 126-128 I noticed from Fig.2 that RSPO4 has higher median (the highest) than LEF1, please comment on this result go into more details.

- Lines 129-132 need a better explanation; is there a Supp. Fig or Table that explain the results of DEG? Please reformulate this sentence and, if it is the case, add the materials necessary to understand it.

- Line 133-136 the authors report that: “the prediction of subtype by LEF1 expression level reached 0.933 (CI: 0.858-1, Figure 2E).” what about the other prediction subtypes? A better explanation of this is required to understand this part.

Minor point:

- Line 60, please insert a reference for all the information reported.

- Lines 63-64, in the sentence “based on the old technology”, please indicate the technology the authors are referring and insert a reference.

- Lines 74-75, it is true that the data do not require Ethical approval but some of them are of public domain upon a written consent (see above).

- Please, reformulate the line 94-95 and indicate the figure to which the authors are referring, is it Fig 1?

- Lines 100-101 require a reference when is reported that the immune phenotype is accepted as definition of ETP.

- Lines 102-103 need to be reformulated it is not clear.

- Line 105, The sentence ”ETP signature were conducted respectively on the T-ALL cohort with 165 patients.” is incomplete, was the study also conducted using the test group of 53? Since this missing part the lines 105-107 are not clear.

- If the authors want to keep the text as in line 122-125 they have to invert the panels in Figure 2, the D has to be after the C.

- Line 138, the citation Zhao et al. is not present as a reference in the bibliography section.

- Line 146, please add a reference to the statement which says LEF1 influences the overall survival rate in Thymoma.

- Please extend Figure 1 caption with a better explanation of the plot and add a legend that explain the meaning of the shaded and not shaded colors.

- The colors in Figure 2C are not consistent with the ones of other panels of Figure 2.

6. PLOS authors have the option to publish the peer review history of their article (what does this mean?). If published, this will include your full peer review and any attached files.

Reviewer #1: No

Reviewer #2: No

---

## [Author Response · Author response to Decision Letter 0]

20 Mar 2020

Reviewer #1: Wang and Zhang report in silicon data pointing towards the differential diagnostic potential of LEF1 expression to detect early precursor T-ALL, in which it is expected to be reduced.

First of all the paper suffers from the at many occasions poor and difficult to understand written English that needs editing.

Thanks for the reviewer’s advice. The revised version is polished thoroughly.

Second, since the paper will primarily be percepted by MD who are not that aware of the biostatistical methods used in silicon, proper citation of papers describing what exactly has been done and its feasibility should be considered.

I am not sure that statements as this given on lines 94-95 are per se deducible from the databases used and are justified.

Thanks for the reviewer’s comments. The evidence which support the statement has been further extended and well explained.

The two paragraphs following lines 94-95 were the evidence to clarify and support the statement. The first paragraph introduces the ETP classification by immune phenotyping which is widely used in clinical setting. The second paragraph describes the classification based on gene expression profile. In the revision, we moved the statement after the two paragraphs. 

To explain in more details, as shown in Figure 1, the classification of ETP by immune-phenotyping (labeled “ETP-status” in Figure 1A and 1B) is inconsistent with that by unsupervised clustering of gene expression profile (labeled “Cluster” and “Cluster-35genes” in Figure 1A and 1B). Each column is a patient and rows represent the classification methods. Dark colors (dark red, orange, and dark blue,) mean the patient is classified to ETP group. To the contrast, light colors (pink, yellow, and light blue) mean the patient is classified to non-ETP group. If the same patient shows dark colors in all three rows, it means the patient is consistently diagnosed as ETP-ALL. However, this is not the case for more than half of the patients. 

If we consider classification by immune phenotyping as a golden standard, then the other two methods have failed in many cases to classify the patients into the correct group.

A more extensive explanation of Figure 1 is added to the legend.

Many additional statements are poorly supported by numeric data in the current paper version. Figure 1 is not telling a lot without more explanatory notes in the legend, actually I was not able to link the data on lines 108-112 to the optics of the figure.

We have extended the figure legend to explain Figure 1 in more details. We hope the reply above provides a better explanation to Figure 1. 

Finally, the results need tissue validation on 5 biopsies of early precursor T-ALL in comparison with 10 more "maturated" cases by means of immunohistochemistry with the nicely working and several times reported antibody.

We appreciate the suggestion about validation in wet lab. Unfortunately, it is not feasible for this study and beyond our scope as being bioinformaticians. Moreover, the lower expression level of LEF1 in ETP comparing to maturated cases has been confirmed in many Microarray studies including GSE28703 (Gutierrez et al. 2011) and GSE8879 (Coustan-Smith et al. 2009).

Coustan-Smith E, Mullighan CG, Onciu M, Behm FG, Raimondi SC, Pei D, et al. Early T-cell precursor leukaemia: a subtype of very high-risk acute lymphoblastic leukaemia. Lancet Oncol. 2009;10(2):147-56.

Gutierrez A, Kentsis A, Sanda T, Holmfeldt L, Chen S-C, Zhang J, et al. The BCL11B tumor suppressor is mutated across the major molecular subtypes of T-cell acute lymphoblastic leukemia. Blood, The Journal of the American Society of Hematology. 2011;118(15):4169-73.

The discussion on the role of LEF1 in tumourigenesis does not add anything to the message of the paper and may be omitted.

In this study, we found that LEF1 could predict the ETP status of T-ALL. This finding indicates that LEF1 may play an important role in the early stage of T-ALL development. The discussion on the role of LEF1 provides an important biological evidence to support our hypothesis derived from bioinformatic analyses. Thus, we maintained the discussion in this revision. 

 

Reviewer #2: Wang and Zhang in this research article investigated, through a clustering approach, if different methods used to diagnose Early T-cell Precursor (ETP) on Acute T-Cell Lymphoblastic Leukemia (T-ALL) are consistent between them. Moreover, they applied a prediction model in order to validate the goodness of LEF1 as molecular biomarker in detecting the status of ETP and therefore stratifying a risk group, moreover they used this approach to find other molecular biomarkers that might be useful to improve the diagnosis of ETP.

They found that the methods used to diagnose ETP not always overlap between them and that LEF1 marker has high accuracy in explaining the ETP status, finally they produced interesting and comparable results to LEF1 using other biomarkers.

The purposes of the study are interesting and well explained, but I was wondering if the authors could give a better explanation I) of the study subject used II) of the computational methods used, and III) of the results obtained. These three points are fundamental to make the study suitable for publication because they will manifest the accuracy of the methodology used and they will give the tools to reproduce the analyses and to understand them.

We appreciate the reviewer’s inputs. Your comments are valuable to our study. We have provided more explanations in this revision for the three aspects suggested. See responses below.

For this reason I would not consider the paper in its present form suitable for publication, but it shows sufficient potential to be reconsidered if it will be substantial revised in the major and minor points as I indicated below.

Major points:

I am suggesting a substantial extension of the method section with a better explanation of the parameters used and the reason why the authors opted for these specific unsupervised clustering and prediction model. I would also suggest to use, beside the one already used, another unsupervised clustering method to validate the results obtained.

Additionally I found some results not properly explained and improvable with some changes in the text and in the figures.

Specifically,

- In the study subjects section, the authors refer to a TARGET T-ALL dataset which includes 165 samples. I suggest to explain the characteristics of this dataset and why the authors preferred this data among the others available. In order to do this the author can report the website and the definition of the acronym already at the beginning of this section.

Additionally, looking at the TARGET publication guidelines of this dataset (https://ocg.cancer.gov/programs/target/target-publication-guidelines) the “TARGET data are available without restrictions on their use in publications or presentations, with the exception of the integrated Acute Lymphoblastic Leukemia (ALL) dataset.” and the “Investigators may only publish an ALL manuscript before the TARGET project team has published their global analysis on that tumor type IF the publication uses a very limited dataset (less than 5 genes) or the author has received written approval from the appropriate TARGET disease project team leaders.”. At this point, since this study is focused on the ALL dataset I was wondering if either this study used only 5 genes or the authors obtained the written approval to use this data. Unfortunately both of these information are missing in the current version of the manuscript.

We agree with the concerns. More information about the datasets is added in section “Study subjects” in the revised version. 

The original study on TARGET T-ALL has been published.

“Liu Y, Easton J, Shao Y, Maciaszek J, Wang Z, Wilkinson MR, McCastlain K, Edmonson M, Pounds SB, Shi L, Zhou X. The genomic landscape of pediatric and young adult T-lineage acute lymphoblastic leukemia. Nature genetics. 2017 Aug;49(8):1211.” 

Hence, the TARGET T-ALL data can be used without restriction in publications. We cited the original study in the revision.

Moreover, in this study 53 individuals has been taken from GEO with the ID GSE42328 but the reference from where these data have been generated which is “Van Vlierberghe P, Ambesi-Impiombato A, De Keersmaecker K, Hadler M et al. Prognostic relevance of integrated genetic profiling in adult T-cell acute lymphoblastic leukemia. Blood 2013 Jul 4;122(1):74-82. PMID: 23687089”, is not indicated, please cite it and add it to the bibliography.

Thanks for pointing that out. The reference is added in revision.

Finally, please report the number of genes (total, median, mean of each of the dataset) and be more specific about the normalization and logarithmic transformation performed.

Those are now reported in the section “Study subjects”, as suggested. 

- In the unsupervised clustering method section it is not reported how the selection of the top 5,000 most variable genes occurred, please add also the reason why the author chose these number, a reference might help.

Additionally, please add a reference that justifies the use of Pearson’s correlation coefficient for “10,000 times to achieve a consensus” or extend this part in order to give the reader a better explanation of the methods employed. As indicated above I also suggest to apply another clustering method to confirm these results.

Selection a subset of the most variable genes ranges from 1,000 to 5,000 has been seen in many studies. In this study, 5,000 genes were selected as recommended by the writer of R package ConsensusClusterPlus. Citation is added.

We agree with the reviewer that unsupervised clustering could generate unstable results. The consensus clustering algorithm proposed by Monti et al. based on resampling could assess the stability of the true clusters, capture the consensus among several clustering runs. The algorithm subsampling a proportion of samples and a proportion of features (genes) each time. The hierarchy clustering was employed in each round of the sub-sample. This process is repeated for 10,000 times. Then, a final agglomerative hierarchical clustering is generated.

Wilkerson MD, Hayes DN. ConsensusClusterPlus: a class discovery tool with confidence assessments and item tracking. Bioinformatics. 2010;26(12):1572-3.

Monti S, Tamayo P, Mesirov J, Golub T. Consensus clustering: a resampling-based method for class discovery and visualization of gene expression microarray data. Mach Learn. 2003;52(1-2):91-118.

- In the prediction model section the authors report that the “80% of the patients were randomly assigned to the training set and rest 20% to the test set”, I find disagreement with the numbers indicated in the previous sections: 53 (test-set in the study subjects section) is not the 20% of 218 (165 of TARGET + 53 of GSE4328); moreover, even if it was, the 20% were not randomly chosen because they belong to the same group/experiment. Please, clarify this and include references and a better explanation of the parameters used in here.

Sorry for the confusing definitions about the training and test datasets. 

Cross-validation was only conducted in the TARGET T-ALL. The purpose of cross-validation is tuning the model parameters and estimating the prediction performance. The TARGET T-ALL dataset was spitted to training set (132, 80% of 165) and test set (33, 20% of 165) for each round of CV.

After the parameters were tuned, the whole TARGET T-ALL was used to fit the final model. Then the model was tested in the GSE42328. That’s why we called the TARGET T-ALL as the “training set”, while the GSE42328 as the “test set” in the submitted version. To avoid this confusion, we defined the GSE42328 dataset as “secondary/external validation dataset” in the revision.

- Line 124-125 I see from the plot that the two highest accuracy values in terms of AUC of ROC are LEF1 and OGN not LE1 and CD5 as indicated in the manuscript, please, clarify this part.

Thanks for pointing that out. The two highest accuracy values should be LEF1 and OGN. 

- Line 126-128 I noticed from Fig.2 that RSPO4 has higher median (the highest) than LEF1, please comment on this result go into more details.

In 94 times out of 100 of CV, LEF1 was selected in the model. This is the most important evidence that LEF1 plays a key role to predict the ETP status. While RSPO4 was selected in 48 out of 100 times. 

The median value of RSPO4’s coefficient in 48 rounds of CV is slightly higher than LEF1’s in 94 rounds. It indicates that RSPO4 can also be a good candidate to predict ETP status, but not as good as LEF1.

- Lines 129-132 need a better explanation; is there a Supp. Fig or Table that explain the results of DEG? Please reformulate this sentence and, if it is the case, add the materials necessary to understand it.

Supp. Table (Supplementary Table 1) and relevant description (Lines 208 to 212, and lines 338-340) on differential expression analysis are added.

- Line 133-136 the authors report that: “the prediction of subtype by LEF1 expression level reached 0.933 (CI: 0.858-1, Figure 2E).” what about the other prediction subtypes? A better explanation of this is required to understand this part.

The “subtype” was referred to ETP status. We revised that sentence as “The prediction of ETP status by LEF1 expression level reached 0.933”.

Minor point:

- Line 60, please insert a reference for all the information reported.

A reference is added.

- Lines 63-64, in the sentence “based on the old technology”, please indicate the technology the authors are referring and insert a reference.

Manuscript is revised to explicitly state “based on the measurement of flow cytometry”

- Lines 74-75, it is true that the data do not require Ethical approval but some of them are of public domain upon a written consent (see above).

As explained above, data used in this study do not require either ethical approval or written consent.

- Please, reformulate the line 94-95 and indicate the figure to which the authors are referring, is it Fig 1?

Thanks for the comment. Taking the two reviewers’ suggestions together, this section has been re-written.

- Lines 100-101 require a reference when is reported that the immune phenotype is accepted as definition of ETP.

A reference is added.

- Lines 102-103 need to be reformulated it is not clear.

This section has been re-written.

- Line 105, The sentence ”ETP signature were conducted respectively on the T-ALL cohort with 165 patients.” is incomplete, was the study also conducted using the test group of 53? Since this missing part the lines 105-107 are not clear.

This section has been re-written.

- If the authors want to keep the text as in line 122-125 they have to invert the panels in Figure 2, the D has to be after the C.

The two figures are switched.

- Line 138, the citation Zhao et al. is not present as a reference in the bibliography section.

The reference is added.

- Line 146, please add a reference to the statement which says LEF1 influences the overall survival rate in Thymoma.

Supporting information is in supplementary table 2, which is added.

- Please extend Figure 1 caption with a better explanation of the plot and add a legend that explain the meaning of the shaded and not shaded colors.

Figure legend and an extended explanation of Figure 1 are added.

- The colors in Figure 2C are not consistent with the ones of other panels of Figure 2.

The colors are adjusted to be consistent with Figure 2B and 2C.

---

## [Decision Letter · Decision Letter 1]

3 Apr 2020

PONE-D-20-01174R1

low LEF1 expression is a biomarker of early T-cell precursor, an aggressive subtype of T-cell lymphoblastic leukemia

PLOS ONE

Dear Dr. Wang,

Thank you for submitting your manuscript to PLOS ONE. After careful consideration, we feel that it has merit but does not fully meet PLOS ONE’s publication criteria as it currently stands. Therefore, we invite you to submit a revised version of the manuscript that addresses the points raised during the review process by both Reviewers.

We would appreciate receiving your revised manuscript by May 17 2020 11:59PM. To enhance the reproducibility of your results, we recommend that if applicable you deposit your laboratory protocols in protocols.io, where a protocol can be assigned its own identifier (DOI) such that it can be cited independently in the future. For instructions see: http://journals.plos.org/plosone/s/submission-guidelines#loc-laboratory-protocols

We look forward to receiving your revised manuscript.

Kind regards,

Francesco Bertolini, MD, PhD

Academic Editor

PLOS ONE

Reviewers' comments:

Reviewer's Responses to Questions

**Comments to the Author**

1. If the authors have adequately addressed your comments raised in a previous round of review and you feel that this manuscript is now acceptable for publication, you may indicate that here to bypass the “Comments to the Author” section, enter your conflict of interest statement in the “Confidential to Editor” section, and submit your "Accept" recommendation.

Reviewer #1: (No Response)

Reviewer #2: (No Response)

2. Is the manuscript technically sound, and do the data support the conclusions?

Reviewer #1: Yes

Reviewer #2: Yes

3. Has the statistical analysis been performed appropriately and rigorously? 

Reviewer #1: I Don't Know

Reviewer #2: Yes

4. Have the authors made all data underlying the findings in their manuscript fully available?

Reviewer #1: Yes

Reviewer #2: Yes

5. Is the manuscript presented in an intelligible fashion and written in standard English?

Reviewer #1: Yes

Reviewer #2: No

6. Review Comments to the Author

Reviewer #1: The authors considerably improved their paper. I am not competent enough to address the biostatistical issues related to the manuscript; here I fully rely on the opinion of referee 2. I still think that the discussion on the role of LEF1 in tumorigenesis, which is textbook knowledge, does not add anything to the message of the paper and should be either shortened or omitted.

Reviewer #2: Wang and Zhang in their revised research article entitled: “low LEF1 expression is a biomarker of early T-cell precursor, an aggressive subtype of T-cell lymphoblastic leukemia” have addressed most of the suggestions raised. However, I think that many minor changes are required to make the manuscript suitable for the publication and I encourage the author to do them. I appreciated the availability of the scripts used.

Minor points above:

In the title please indicate the first letter in upper case.

Line 68 reformulate the sentence, indeed “whole transcriptome and by the 35-gene ETP signature were conducted respectively on...” may be understood as one analysis was applied on TARGET while the other on GSE42328 instead both the analyses were applied to both the data sets.

Lines 109-110 explain the concept of specificity and sensitivity and how you calculated them adding few sentences on the methods, maybe in the “Other Statistic section”.

Lines 154-163 please substitute the word “study” with synonyms.

If, from TCGA dataset, the authors used RNA sequencing data, it would be useful to add the number of genes (if specified) for completeness.

Unsupervised clustering section: the authors have deleted the selection of the 5,000 genes which was reported in the previous version of the manuscript; in my opinion it is important to add it back in order to let the reader fully understand the method used to select these genes.

Supervised prediction section, please add a reference in the first two sentences of this paragraph. Moreover, please provide the numbers and parameters used as pointed out in the answer to the reviewer. Finally I do not see along the text that the GSE42328 data set is called neither “external” nor “secondary” (as indicated in the answer to reviewers) but only “validation”, is it intended?

Please, add in the m&m, under the section “Other Statistics”, the acronym DEGs when referring to the explanation of this methodolgy. Moreover, please add a reference and the method used to do the Cox regression and add the “survival” R library (with its citation) if it was used to calculate the survival rate.

Supplementary Tables and Materials need captions (legend) to explain what they show. I suggest merging all the tables in a unique .xlsx file with different sheets and then add a caption that explains all the sheets (Tables).

Additionally, Supplementary Table 2 has the name of the sheet which is "Table_S1", please correct it and also indicate which threshold you used when the Wilcoxon test was performed and what the data on the columns mean.

The same for Supplementary Table 3, in which the name sheet is mislabelled “Table_S2_LEF1_TCGA_survival” and there is no caption.

Please adjust the references according to the PLOS ONE format.

7. PLOS authors have the option to publish the peer review history of their article (what does this mean?). If published, this will include your full peer review and any attached files.

Reviewer #1: No

Reviewer #2: No

---

## [Author Response · Author response to Decision Letter 1]

5 Apr 2020

Thanks for the reviewers’ advice. The manuscript was edited accordingly.

Reviewer #1: The authors considerably improved their paper. I am not competent enough to address the biostatistical issues related to the manuscript; here I fully rely on the opinion of referee 2. I still think that the discussion on the role of LEF1 in tumorigenesis, which is textbook knowledge, does not add anything to the message of the paper and should be either shortened or omitted.

Thanks for reviewing our manuscript. Considering the reviewer's suggestion, we tried to shorten the discussion on LEF1's biological function, but found that each sentence is supporting our hypothesis and point of view. In addition, the description and discussion based on re-analyzing the TCGA Pancancer data is our original work, not referred from any textbook or other publications. We prefer to keep them. But if the reviewer has a more specific suggestion, e.g. removing a specific sentence, we could try to revise this section again.

Reviewer #2: Wang and Zhang in their revised research article entitled: “low LEF1 expression is a biomarker of early T-cell precursor, an aggressive subtype of T-cell lymphoblastic leukemia” have addressed most of the suggestions raised. However, I think that many minor changes are required to make the manuscript suitable for the publication and I encourage the author to do them. I appreciated the availability of the scripts used.

Minor points above:

In the title please indicate the first letter in upper case.

Corrected.

Line 68 reformulate the sentence, indeed “whole transcriptome and by the 35-gene ETP signature were conducted respectively on...” may be understood as one analysis was applied on TARGET while the other on GSE42328 instead both the analyses were applied to both the data sets.

“respectively” is deleted.

Lines 109-110 explain the concept of specificity and sensitivity and how you calculated them adding few sentences on the methods, maybe in the “Other Statistic section”.

Explained as suggested.

Lines 154-163 please substitute the word “study” with synonyms.

“study” is replaced with “publication”.

If, from TCGA dataset, the authors used RNA sequencing data, it would be useful to add the number of genes (if specified) for completeness.

We only used the expression level of LEF1 gene from the RNA sequencing dataset.

Unsupervised clustering section: the authors have deleted the selection of the 5,000 genes which was reported in the previous version of the manuscript; in my opinion it is important to add it back in order to let the reader fully understand the method used to select these genes.

Thanks for reminding. It was removed by mistake. That sentence is added back in this revision.

Supervised prediction section, please add a reference in the first two sentences of this paragraph. Moreover, please provide the numbers and parameters used as pointed out in the answer to the reviewer. Finally I do not see along the text that the GSE42328 data set is called neither “external” nor “secondary” (as indicated in the answer to reviewers) but only “validation”, is it intended?

Reference is added and numbers are provided.

To be consistent and concise, we decided to call the GSE42328 the validation dataset.

Please, add in the m&m, under the section “Other Statistics”, the acronym DEGs when referring to the explanation of this methodolgy. Moreover, please add a reference and the method used to do the Cox regression and add the “survival” R library (with its citation) if it was used to calculate the survival rate.

Added as suggested.

Supplementary Tables and Materials need captions (legend) to explain what they show. I suggest merging all the tables in a unique .xlsx file with different sheets and then add a caption that explains all the sheets (Tables).

Additionally, Supplementary Table 2 has the name of the sheet which is "Table_S1", please correct it and also indicate which threshold you used when the Wilcoxon test was performed and what the data on the columns mean.

The same for Supplementary Table 3, in which the name sheet is mislabelled “Table_S2_LEF1_TCGA_survival” and there is no caption.

Corrected as suggested.

Please adjust the references according to the PLOS ONE format.

Done.

---

## [Decision Letter · Decision Letter 2]

17 Apr 2020

low LEF1 expression is a biomarker of early T-cell precursor, an aggressive subtype of T-cell lymphoblastic leukemia

PONE-D-20-01174R2

Dear Dr. Wang,

We are pleased to inform you that your manuscript has been judged scientifically suitable for publication and will be formally accepted for publication once it complies with all outstanding technical requirements.

With kind regards,

Francesco Bertolini, MD, PhD

Academic Editor

PLOS ONE

Additional Editor Comments (optional):

Reviewers' comments:

Reviewer's Responses to Questions

**Comments to the Author**

1. If the authors have adequately addressed your comments raised in a previous round of review and you feel that this manuscript is now acceptable for publication, you may indicate that here to bypass the “Comments to the Author” section, enter your conflict of interest statement in the “Confidential to Editor” section, and submit your "Accept" recommendation.

Reviewer #1: All comments have been addressed

Reviewer #2: (No Response)

2. Is the manuscript technically sound, and do the data support the conclusions?

Reviewer #1: Yes

Reviewer #2: Yes

3. Has the statistical analysis been performed appropriately and rigorously? 

Reviewer #1: Yes

Reviewer #2: Yes

4. Have the authors made all data underlying the findings in their manuscript fully available?

Reviewer #1: Yes

Reviewer #2: Yes

5. Is the manuscript presented in an intelligible fashion and written in standard English?

Reviewer #1: Yes

Reviewer #2: Yes

6. Review Comments to the Author

Reviewer #1: I have no additional comments. The authors have addressed my points and I accept their arguments with respect to keeping the discussion on the bilogical functions of LEF1.

Reviewer #2: Wang and Zhang revised and improved their study according to the suggestions of the reviewers. The paper, in my opinion, is now suitable for publication after the correction of some typo as:

-The number with more than 3 digits should be indicated as XX,XXX

-Line 202: please correct the name of the test in “non-parametric Wilcoxon”

-Please adjust the references according to the PLOS ONE format, I see more than one doi in some of the references and at the ref. number 19 it is missing the last page “2016;128(22):5083-.”, if the article is only of one page I think it should be indicate as 5083.

7. PLOS authors have the option to publish the peer review history of their article (what does this mean?). If published, this will include your full peer review and any attached files.

Reviewer #1: No

Reviewer #2: No

---

## [Editor Report · Acceptance letter]

1 May 2020

PONE-D-20-01174R2 

low LEF1 expression is a biomarker of early T-cell precursor, an aggressive subtype of T-cell lymphoblastic leukemia 

Dear Dr. Wang:

I am pleased to inform you that your manuscript has been deemed suitable for publication in PLOS ONE. Congratulations! Your manuscript is now with our production department. 

With kind regards,

on behalf of

Dr. Francesco Bertolini 

Academic Editor

PLOS ONE